# Context Quality Matters in Training Fusion-in-Decoder for Extractive Open-Domain Question Answering

**Kosuke Akimoto** and **Kunihiro Takeoka** and **Masafumi Oyamada**
NEC Corporation
{kosuke_a, k_takeoka, oyamada}@nec.com

## Abstract

Retrieval-augmented generation models augment knowledge encoded in a language model by providing additional relevant external knowledge (context) during generation. Although it has been shown that the quantity and quality of context impact the performance of retrieval-augmented generation models during inference, limited research explores how these characteristics affect model training. This paper explores how context quantity and quality during model training affect the performance of Fusion-in-Decoder (FiD), the state-of-the-art retrieval-augmented generation model, in extractive open-domain question answering tasks. Experimental results suggest that FiD models overfit to context quality during training and show suboptimal performance when evaluated on different context quality. Through the experimental results, we also reveal FiD models trained with different context quality have different cross-attention distribution patterns. Specifically, as context quality during training increases, FiD models tend to attend more uniformly to each passage in context. Finally, based on these observations, we propose a method to mitigate overfitting to specific context quality by introducing bias to the cross-attention distribution, which we demonstrate to be effective in improving the performance of FiD models on different context quality.

## 1 Introduction

Recently, large-scale pre-trained language models have achieved impressive performance in the field of Natural Language Generation, which includes tasks that require real-world knowledge, e.g., closed-book question answering and common sense reasoning (Brown et al., 2020). However, these models are still prone to generate factually incorrect outputs known as hallucinations (Ji et al., 2023), particularly when dealing with rare entities (Mallen et al., 2023). Also, they cannot handle new information that arises after their training phase (Kasai et al., 2022).

In order to address these challenges, retrieval-augmented generation models have been recently proposed (Izacard and Grave, 2021b; Lewis et al., 2020). These models draw inspiration from retrieval-based extractive open-domain question answering methods (Chen et al., 2017) and utilize additional relevant external knowledge (e.g., a Wikipedia article about an entity in a given question) during generation to augment knowledge encoded in a language model. Retrieval-augmented generation models have demonstrated effectiveness in knowledge-intensive tasks (Petroni et al., 2021) such as question answering and fact checking (Hofstätter et al., 2022), and have been reported to reduce hallucinations in dialogue tasks (Shuster et al., 2021).

The external knowledge given to the models is called a *context*, and it is usually obtained through information retrieval systems (Lin et al., 2022). Multiple passages, typically up to 100, are often used collectively as a single context to ensure the high recall of relevant information. This strategy addresses the limitations of retrieval systems, which may return irrelevant passages and fail to capture relevant information in the top results. When dealing with contexts composed of multiple passages, we can define their *quantity* (the number of passages in the context) and *quality* (the proportion of relevant passages in the context). Since the context quantity and quality vary depending on model configuration or application, e.g., the performance of the retrieval system and the computational resources available, understanding how these characteristics impact the model performance becomes an important research question.

Indeed, during the inference phase, it has been shown that the quantity and quality of contexts impact the performance of retrieval-augmented generation models. For example, Izacard and Grave

(2021b) showed that increasing the number of top-ranked retrieved passages used as a context during inference improves the performance of their model in the question answering task, and Weller et al. (2022) found that the model prediction is distracted more strongly as the proportion of conflicting misinformation in the context increases.

However, regarding the training phase, it is not yet fully understood how these context characteristics impact the performance of the trained models. Limited research suggests that increasing the number of retrieved passages used as a context during training improves question answering performance (Izacard and Grave, 2021b) and reduces memorization (Chen et al., 2022). Still, the impact of quantity and quality of contexts is mixed in these studies, as relevant passages are typically biased towards higher rank in the retrieval result, and simply increasing the number of top-ranked passages changes both the quantity and quality of the context.

In this paper, we focused on extractive open-domain question answering tasks and investigated the impact of context quantity and quality on the training of Fusion-in-Decoder (FiD) (Izacard and Grave, 2021b), a state-of-the-art retrieval-augmented generation model. We demonstrate that context quality during training affects the performance of the trained model. As far as our knowledge, this work is the first attempt to explicitly control context quality and investigate its effect on training of retrieval-augmented generation models.

Key insights obtained through our experiments are as follows:

- FiD models overfit to context quality during training, resulting in deteriorated performance when evaluated on a different quality of context.

- FiD models overfit less to context quantity compared to context quality.

- FiD models trained with different context qualities show different patterns of cross-attention probability. As context quality during training increases, the trained models tend to attend more uniformly to each passage in context and vice versa.

Based on these observations, we propose a method to mitigate the overfitting of a trained FiD model to specific context quality without additional training by controlling the selectivity of its cross-attention distribution. We present an empirical analysis demonstrating the proposed method's effectiveness in improving the performance of a trained FiD model when deployed in environments with different context quality than those used during training.

## 2 Experimental Setup

In this section, we describe the task (§2.1) and model architecture (§2.2) used in our experiments, and we define quality and quantity that we used in this paper (§2.3, 2.4).

### 2.1 Task and Dataset

This study focuses on the extractive open-domain question answering task, where models have to extract answers from retrieved documents. We conducted experiments on two standard benchmark datasets of the task:

- **Natural Questions** (Kwiatkowski et al., 2019) contains questions submitted to Google Search engine. We use the open-domain version of this dataset presented by Lee et al. (2019).

- **TriviaQA** (Joshi et al., 2017) contains questions authored by trivia enthusiasts. Following Lee et al. (2019), we use the unfiltered set of the dataset.

For each dataset, following (Izacard and Grave, 2021b), we used top-100 passages retrieved by DPR (Karpukhin et al., 2020)[1]. As an evaluation metric, we computed the exact match (EM) between a ground-truth answer and predicted answer generated by greedy decoding[2]. We evaluated the performance on the development set of each dataset.

### 2.2 Model

In our experiments, we focused on **Fusion-in-Decoder (FiD)** (Izacard and Grave, 2021b), a state-of-the-art architecture for retrieval-augmented generation model. FiD is extended from sequence-to-sequence models, such as T5 (Raffel et al., 2020), and consists of a Transformer encoder $E$ and decoder $D$ (Vaswani et al., 2017).

---

[1] The retrieved passages were obtained from the published dataset at https://github.com/facebookresearch/FiD.

[2] If a question has multiple ground-truth answers, EM is computed as one if the predicted answer matches any one of the ground-truth answers.

Given a question $q$ and its context $c = \{p_i\}_{i=1}^N$, where $p_i$ is the $i$-th passage of the context, a FiD model converts each passage $p_i$ to $\tilde{p}_i$ by a template `"question: {q} title: {t} context: {c}"`. Here, `{q}`, `{t}`, and `{c}` are respectively replaced by $q$, the title of $p_i$, and the main text of $p_i$. Then, a FiD model independently encodes each converted passage $\tilde{p}_i$ by the encoder $E$ and feeds their concatenation to the decoder $D$ to get predicted answer $a$ as follows:

$$a = D([E(\tilde{p}_1); ...; E(\tilde{p}_n)]). \tag{1}$$

We followed standard practice and trained FiD models by minimizing a cross-entropy loss of a ground-truth answer. As the position of a passage is not considered while encoding, the prediction of a FiD model is insensitive to the order of the passages. Thus we did not perform any special shuffling or ordering of the passages during training and evaluation. We used `t5-base`[3] (Raffel et al., 2020) to initialize the model. See Appendix A for other implementation details of training and inference of FiD models.

## 2.3 Relevant and Irrelevant Passage

In this paper, we adopt the same definition of *relevant* and *irrelevant* passage as in Li et al. (2022). More specifically, a passage is relevant to a question if it logically entails an answer to the question, and it is irrelevant if it does not.

However, in our open-domain setting, no ground-truth annotation of passage relevance exists for Natural Questions and TriviaQA. As discussed by Li et al. (2022), a simple rule that determines a passage that contains a ground-truth answer as a relevant passage is insufficient to filter out irrelevant passages that contain the answer but do not entail the answer. Since accurately determining whether a passage is relevant or not is crucial for estimating context quality, we applied an additional rule to extract relevant passages. We fed a pair of a question and a passage to a pre-trained question answering model[4], and deemed the passage relevant if the predicted answer matched a ground-truth answer to the question[5]. Following Li et al. (2022),

we considered a passage that did not contain any ground-truth answers to the question as irrelevant.

We respectively denote the set of relevant and irrelevant passages of question $q$ by $R(q)$ and $\bar{R}(q)$, and we omit $(q)$ when it is not necessary.

## 2.4 Context Quality and Quantity

For a question $q$ and a context $c = \{p_i\}_{i=1}^N$ of $N$ passages, we define context quality and quantity as follows:

- **Context quality** is defined as the proportion of passages in $c$ that is *relevant* to $q$, i.e. $\frac{|R(q)|}{N}$.

- **Context quantity** is the number of passages in $c$, i.e. $N$.

## 3 Case Studies

In this section, we describe our experiments to investigate how context quantity and quality during training affect the performance of FiD models. Throughout the experiments in this section, we created various training and evaluation environments with controlled context quantity or quality by sampling $n^+$ relevant passages from $R(q)$ and $n^-$ irrelevant passages from $\bar{R}(q)$ for each question $q$, without replacement[6]. In the rest of this paper, we define $n = n^+ + n^-$ as the number of total passages and $k = \frac{n^-}{n^+}$ as the ratio of irrelevant passages to relevant ones. We will use subscripts $\cdot_{\text{train}}$ and $\cdot_{\text{eval}}$ respectively to denote the values of training and evaluation environments if required.

## 3.1 Effect of Context Quality during Training

**Setting:** To investigate the effect of context quality during model training, we created training and evaluation environments with the same context quantity but different context qualities. More specifically, for each total number of passages (i.e., context quantity) $n$ in $\{10, 25, 60\}$, we varied the value of $n^+$ among $\{1, 2, 3\}$ (Natural Question) or $\{1, 2, 3, 5, 10\}$ (TriviaQA) to obtain environments with different context qualities.

**Result:** Figure 1 shows the performance of models with different training context qualities[7]. We can obtain the following observations from the figure: (i) For a given evaluation context quality, models trained with similar context quality showed the

---

[3] https://huggingface.co/t5-base
[4] To annotate a passage of a question $q$ in dataset $\mathcal{D}$, we used FiD models trained on a subset of $\mathcal{D}$ that did not contain $q$. See Appendix C for more details.
[5] We discarded those passages that contained a ground-truth answer but did not pass the additional filter, and we did not use those passage in our experiments.

[6] See Appendix B for more details of the experimental design.
[7] See Figure 6,7 and Table 7,8 in Appendix D for full results.

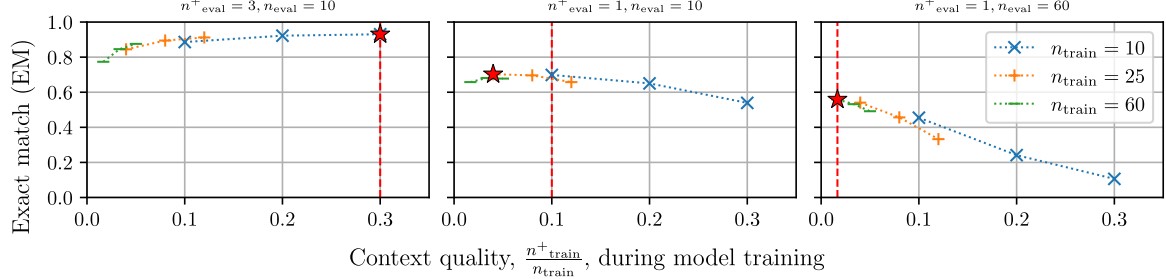

Figure 1: Performance of FiD models on Natural Questions with varying training context quality. Panels represent different evaluation environments with different $(n_{eval}^+, n_{eval})$ pairs, and a red dashed line shows the context quality of the corresponding evaluation environment. Red stars represent the best-performed models in the corresponding evaluation environments. Dotted lines show models trained on the same context quantity $n_{train}$.

highest performance. (ii) There was a trend of monotonically decreasing performance as training context quality deviated further from evaluation context quality. (iii) Difference of context quantity had a negligible impact in the above trends.

**Insight:** FiD models overfit to context quality during training, and suboptimal performance can be obtained when evaluated on different context qualities.

### 3.2 Effect of Context Quantity during Training

**Setting:** To investigate the effect of context quantity during model training, we created training and evaluation environments with the same context quality but different context quantities. More specifically, for each ratio $k$ among $\{1, 5, 20\}$, we varied the value of $n^+$ among $\{1, 2, 3\}$ (Natural Questions) or $\{1, 2, 3, 5, 8, 10\}$ (TriviaQA) to change context quantity[8].

**Result:** Figure 2 shows the performance of models with different training context quantities[9]. As can be seen in the figure, the influence of context quantity on model training was generally less significant than that of context quality (§3.1). However, we observed a more significant influence for smaller $k_{train}$ (higher context quality) in comparison to larger $k_{train}$ (lower context quality), especially in the cases where the training context quantity was small. One possible explanation for this behavior is that the impact of noise in the annotation of relevant (or irrelevant) passages disturbs the actual context quality, and this effect is magni-

fied in such cases due to limited context quantities. Nevertheless, our experiments did not reveal a consistent trend in performance changes due to varying context quantity. Hence, the results indicate that context quantity's influence is relatively insignificant compared to context quality's.

**Insight:** Training context quantity has less influence on model performance compared to context quality.

### 3.3 Effect of Mixed Context Quality during Training

Generally, in practical uncontrolled settings, a training dataset for FiD models may consist of questions with different context qualities. Thus, we conducted experiments with mixed context qualities to investigate whether a similar overfitting phenomenon occurs for a training dataset with multiple context qualities.

**Setting:** We created three environments for each dataset. Specifically, context quantity was set to $n = 10$, and $n^+$ was varied among $\{1, 2, 3\}$ for Natural Questions, and context quantity was set to $n = 25$, and $n^+$ was varied among $\{2, 5, 10\}$ for TriviaQA. Then, we uniformly mixed each subset of the three environments and trained FiD models in each of them[10]. Performance in a mixed environment was computed by averaging the performance in each constituting environment.

**Result:** Table 1 shows model performance for each pair of training and evaluation environment in Natural Questions[11]. High scores at diagonal elements of the table show that the models performed best or as well as the best model when they were

---

[8]Since target questions were only guaranteed to have at least 64 irrelevant passages, we did not conduct experiments in environments with $kn^+ > 64$.

[9]See Figure 8, 9 and Table 9, 10 in Appendix D for full results.

[10]We mixed environments by randomly selectig the value of $n^+$ and sampling passages accordingly for each question and training step.

[11]See Table 5 for the result in TriviaQA.

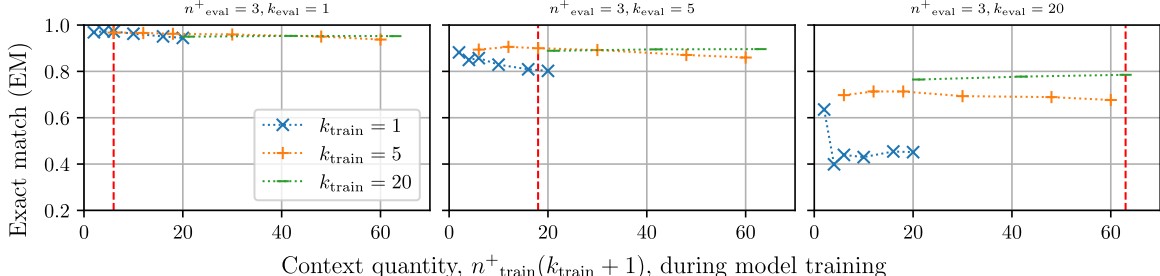

Figure 2: Performance of FiD models on TriviaQA with varying training context quantity. Panels represent different evaluation environments with different $(n^+_{\text{eval}}, k_{\text{eval}})$ pairs, and a red dashed line shows the context quantity of the corresponding evaluation environment. Dotted lines show models trained on the same context quality $\frac{1}{1+k_{\text{train}}}$.

Table 1: Performance of FiD models trained in each mixture of environments on Natural Questions. The color intensity indicates the relative performance within the same evaluation mixture of environments (column). Checkmarks indicate which environment included in each mixture.

| $n^+_{\text{train}}(\downarrow)$ 1 2 3 | $n^+_{\text{eval}}(\rightarrow)$ 1 ✓ | 2 ✓ | 3 | ✓ | ✓ | ✓ | ✓ |
|---|---|---|---|---|---|---|---|
| ✓ | 69.8 | 82.9 | 88.6 | 85.7 | 79.2 | 76.4 | 80.4 |
| ✓ | 65.0 | 85.5 | 92.2 | 88.9 | 78.6 | 75.3 | 80.9 |
| ✓ | 53.9 | 83.4 | 93.0 | 88.2 | 73.5 | 68.7 | 76.8 |
| ✓ ✓ | 62.3 | 85.2 | 92.5 | 88.8 | 77.4 | 73.7 | 80.0 |
| ✓ ✓ | 69.1 | 84.0 | 90.0 | 87.0 | 79.5 | 76.5 | 81.0 |
| ✓ ✓ | 69.7 | 83.9 | 89.6 | 86.8 | 79.6 | 76.8 | 81.1 |
| ✓ ✓ ✓ | 68.3 | 84.6 | 90.7 | 87.6 | 79.5 | 76.4 | 81.2 |

evaluated in the same mixture of environments as one in training. For example, the models trained in the uniform mixture of all environments performed best only when they were evaluated on the same mixture. It suggests that covering all context qualities during evaluation is insufficient for optimal performance, and the distribution of the context qualities also matters to the performance.

**Insight:** FiD models overfit to the distribution of context qualities during training.

### 3.4 Effect of Context Quality during Training on Model's Cross-attention

As we discussed in §3.1, FiDs trained on different context qualities may overfit to each quality, and they perform differently in the same evaluation environment. We hypothesize that overfitting to different context quality occurs due to changes in how a model selects relevant passages since lower context quality may force the model to concentrate more on selecting passages and vice versa. Thus, as a first step to investigate which aspect of model inference is affected by different context qualities during training, we analyzed how patterns

of cross-attention varied among FiD models trained on different context qualities (§3.4.1). Then, we conducted intervention experiments to validate that different patterns of cross-attention explain part of the overfitting to context quality (§3.4.2).

#### 3.4.1 Investigation on Patterns of Cross-attention Probability

**Setting:** We denote cross-attention probability from the $l$-th decoder token to the $j$-th token of the $i$-th input passage $\tilde{p}_i$ at the $k$-th decoder layer by $c^{(k)}_{ijl}$. Following (Izacard and Grave, 2021a), we computed cross-attention probability from the first decoder token, $c^{(k)}_{ij1}$, and we computed aggregated cross-attention probability $\tilde{c}^{(k)}_i = \sum_j c^{(k)}_{ij1}$ for each passage $\tilde{p}_i$.

We conducted the following two analyses:

(i) We analyzed how much cross-attention probability was allocated to relevant passages at each layer, i.e., $\sum_{i \in \{i | p_i \in R\}} \tilde{c}^{(k)}_i$.

(ii) We analyzed the difference between the distribution of cross-attention probability to relevant passages, i.e., $\{\tilde{c}^{(k)}_i | i \in R\}$, and that to irrelevant passages, i.e., $\{\tilde{c}^{(k)}_i | i \in \bar{R}\}$.

We focused our analyses on FiD models trained for Natural Questions in §3.1 with the following settings: $(n, n^+) \in \{(10, 1), (10, 2), (10, 3)\}$. Note that these models were trained with the same context quantity but different qualities. We analyzed these models in two evaluation environments of Natural Questions with the following settings: $(n^+, n^-) \in \{(3, 7), (3, 57)\}$.

**Result:** Table 2 shows the cross-attention probability that was allocated to relevant passages at each layer (Analysis (i)). As shown in the table, in both evaluation environments, FiD models trained

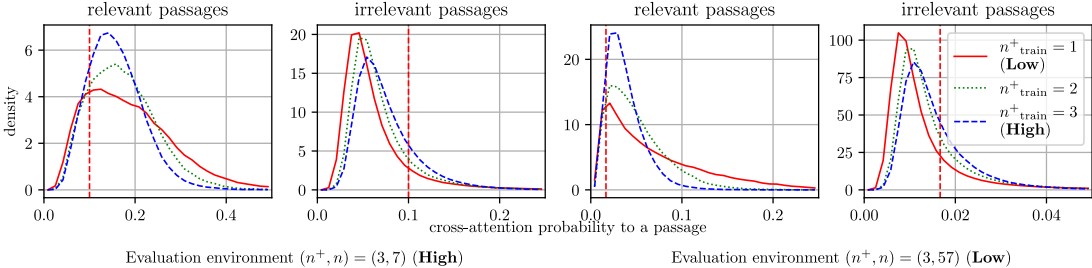

Evaluation environment $(n^+, n) = (3, 7)$ (**High**)  Evaluation environment $(n^+, n) = (3, 57)$ (**Low**)

Figure 3: Distribution of cross-attention probability to each relevant or irrelevant passage at Layer 9. A similar trend can be seen in other higher layers. Red vertical dashed lines represent uniform cross-attention probability, i.e., $\frac{1}{N}$ if context quantity is $N$.

Table 2: Cross-attention probability allocated to relevant passages at each layer. **High** and **Low** respectively represent high and low context quality.

| | Evaluation environment $(n^+, n^-)$ | | | | | |
| | (3, 7) | | | (3, 57) | | |
| | **High** | | | **Low** | | |
| | Training environment $(n, n^+) = (10, n^+)$ | | | | | |
| $n^+$ | 3 | 2 | 1 | 3 | 2 | 1 |
| | **High** | | **Low** | **High** | | **Low** |
| Layer 1 | 31.5 | 31.9 | 31.0 | 5.4 | 5.5 | 5.3 |
| Layer 2 | 31.8 | 32.3 | 32.1 | 5.4 | 5.6 | 5.6 |
| Layer 3 | 33.7 | 34.5 | 33.6 | 6.1 | 6.3 | 6.0 |
| Layer 4 | 32.5 | 33.4 | 32.3 | 5.7 | 6.0 | 5.7 |
| Layer 5 | 32.4 | 33.7 | 33.3 | 5.6 | 6.0 | 6.0 |
| Layer 6 | 33.9 | 35.4 | 36.2 | 6.1 | 6.6 | 7.2 |
| Layer 7 | 41.1 | 45.6 | 47.4 | 8.5 | 10.7 | 13.7 |
| Layer 8 | 40.8 | 44.1 | 45.8 | 8.6 | 10.3 | 13.2 |
| Layer 9 | 47.4 | 51.9 | 56.3 | 11.3 | 15.1 | 21.8 |
| Layer 10 | 48.2 | 52.9 | 55.5 | 10.8 | 13.9 | 20.4 |
| Layer 11 | 45.6 | 49.8 | 53.4 | 10.1 | 12.6 | 19.0 |
| Layer 12 | 39.1 | 41.2 | 42.9 | 9.5 | 11.4 | 15.7 |

with lower context quality attended more strongly to relevant passages, especially at higher layers that are closer to the output layer[12].

A similar trend is also observed in Figure 3 that shows the distribution of cross-attention probability to a relevant or irrelevant passage for each model (Analysis (ii)). Models trained with lower context quality showed more long-tailed distribution for relevant passages, and there was a more significant difference between the distribution for relevant and irrelevant passages, which suggests they are trained to attend more selectively to relevant passages. On the contrary, the distribution is relatively closer to uniform distribution for models trained with higher context quality.

We conjecture that this excessive selectivity of models trained in a low-quality environment may

explain their relatively lower performance in a high-quality environment (§3.2), because such excessive selectivity makes the model overlook necessary information in ignored relevant passages and, as a result, fail to correctly answer the questions. It may be the case that, when evaluated in a high-quality environment (i.e., where the majority of passages are relevant to the question), it is more optimal for the model to examine all passages more uniformly without being overly selective. This claim is empirically supported by the result of our experiments in §4.2.

**Insight:** FiD models trained with different context quality show different levels of selectivity w.r.t. allocation of cross-attention probability. Models trained with lower context quality attend more selectively to relevant passages.

### 3.4.2 Intervention Experiment

Results in §3.4.1 suggests that overfitting of FiD models to different context quality is due to different level of selectivity of their cross-attention. We validate this claim by intervening on the cross-attention probability of these models during inference.

**Setting:** We intervened on the cross-attention probability of FiD models so that the ratio of cross-attention probability to a relevant passage $p_i \in R$ and an irrelevant passage $p_j \in \bar{R}$ to be $r$ for all layers. Intuitively, the model completely ignores irrelevant passages when $r = 0$, whereas the model attends uniformly to all passages when $r = 1$. More specifically, for each decoder layer $k$, we converted original cross-attention probability $c_{ijl}^{(k)}$ into intervened version $c'^{(k)}_{ijl}$ as follows:

$$c'^{(k)}_{ijl} = \frac{w_i^{(k)}}{\sum_{j'} c_{ij'l}^{(k)}} c_{ijl}^{(k)}, \qquad (2)$$

---

[12]The result may also suggest that a function to select relevant passages is learned at higher layers, which are consistent with the claims made by Tenney et al. (2019) that higher layers capture task-specific information.

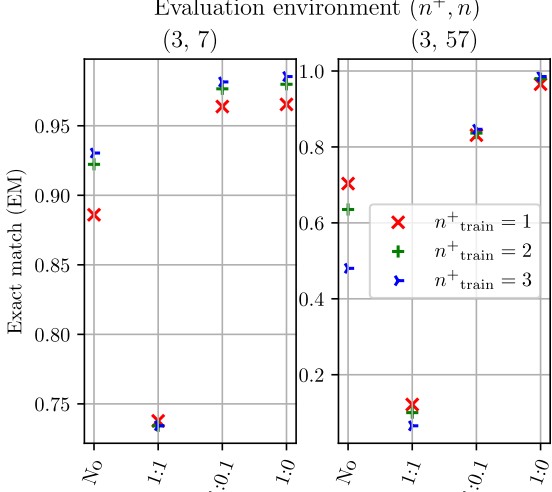

Evaluation environment $(n^+, n)$

Ratio of intervened cross-attention probability
to a relevant and irrelevant passage, $1 : r$

Figure 4: Model performance under intervention on cross-attention probability. "No" represents a setting without intervention.

where $w_i^{(k)} = \frac{1}{n^+ + rn^-}$ if $p_i \in R$ and $w_i^{(k)} = \frac{r}{n^+ + rn^-}$ if $p_i \in \bar{R}$. We selected $r$ from $\{1, 0.1, 0\}$ and conducted experiments in the same models and evaluation environments as in §3.4.1.

**Result:** Figure 4 shows model performance with and without intervention on cross-attention probability. In both evaluation environments with lower and higher context quality, the difference in the performance of the models decreased, and the intervention mitigated the effect of overfitting to context quality.

**Insight:** The result suggests that the difference of cross-attention probability as described in §3.4.1 is one element that explains the overfitting of FiD models to context quality.

## 4 Adapting Models to Different Context Quality

While FiD models overfit to context quality during training as shown in §3, it is not desirable to train a dedicated model for each target environment that has potentially different context qualities from each other. Thus, in this section, we propose a method to mitigate the effect of overfitting and adapt an already trained FiD model to an environment with different context quality.

### 4.1 Proposed Method

Based on the insights in §3.4 that shows the overfitting to context quality occurs due to the differ-

ent levels of selectivity to relevant passages, we propose to change *sharpness* of distribution of cross-attention probability during inference. More specifically, we introduce temperature parameter $T$ ($T > 0$) and compute total cross-attention probability from the $l$-th decoder token to the $i$-th passage at the $k$-th layer as follows:

$$w_{il}^{(k)} = \left( \text{softmax} \left[ \frac{\log(\sum_j c_{1jl})}{T}, ..., \frac{\log(\sum_j c_{Njl})}{T} \right] \right)_i. \tag{3}$$

Then, we use Equation (2) to convert cross-attention probability as in §3.4.2[13]. Intuitively, the model attends more uniformly as $T$ becomes larger, which simulates the overfitting effect of a FiD model trained with higher context quality and vice versa.

Note that our proposed temperature parameter does not change the set of input passages and can be tuned complementary with other existing hyperparameter that changes the set of input passages, e.g., the number of input passages.

### 4.2 Experiment

To validate the effectiveness of our proposed method, we adapted models trained in §3.1 by the proposed method and evaluated their performance on evaluation environments with different context qualities where $n_{\text{eval}}^+ = 3$ and $n_{\text{eval}} \in \{10, 25, 60\}$. Since the temperature parameter $T$ has to be tuned, we conducted 2-fold cross-validation. Specifically, we split the evaluation data into two folds and searched optimal temperature parameter $T^* \in \{0.125, 0.25, 0.5, 1, 2, 4, 8\}$ based on the EM score on one fold, and then, we used a model adapted with $T^*$ to evaluate performance on the other fold[14].

Figure 5 shows the performance of FiD models with and without adaptation by the proposed method[15]. As shown in the figure, the proposed method improved the performance of the models in environments with different context qualities compared to those during training, and it reduced the effect of overfitting to context quality. Also, $T^*$

---

[13]We use $w_{il}^{(k)}$ instead of $w_i^{(k)}$ in Equation (2) when decoding the $l$-th token. Note that the cross-attention probability is sequentially computed for each decoder token during inference.

[14]We used the single temperature parameter $T^*$ for all questions in a fold instead of using a different temperature parameter for each question. Predicting the optimal temperature parameter for each input question is an interesting direction of future works.

[15]See Figure 10 in Appendix D for the result in TriviaQA.

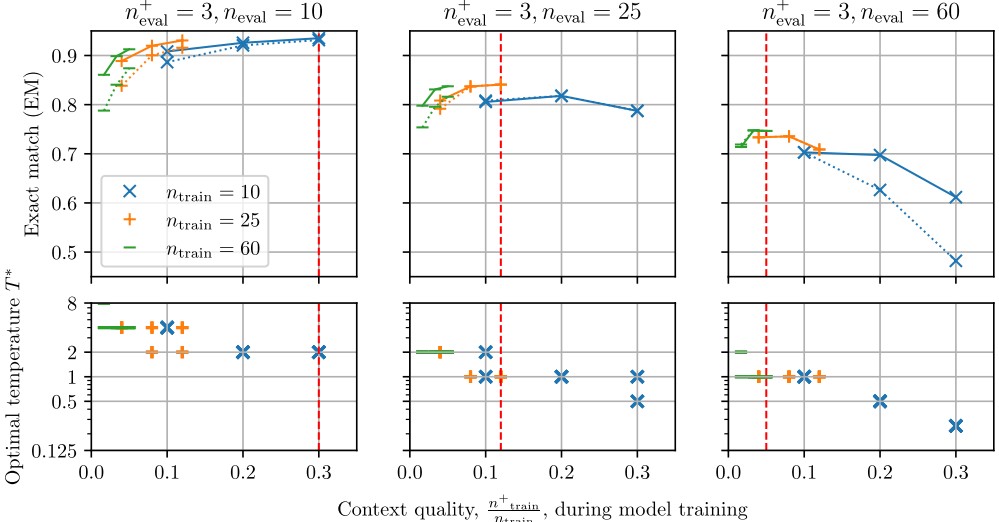

Figure 5: **Top panels:** Performance of FiD models on Natural Questions **with** adaptation by the proposed method (solid lines) and **without** adaptation (dotted lines). **Bottom panels:** Optimal temperature parameter $T^*$ selected for each model. Multiple $T^*$ were selected for some context qualities, i.e., training environments, because we selected $T^*$ for each of the three models trained with different random seeds for each training environment.
Panels represent different evaluation environments with different $(n_{\text{eval}}^+, n_{\text{eval}})$ pairs, and a red dashed line shows the context quality of the corresponding evaluation environment.

increased in the case of lower context qualities during training, and vice versa, which corroborates our finding that more uniform cross-attention, corresponding to higher $T^*$, is effective when context quality in evaluation is higher than one in training.

## 5 Related Works

**Retrieval-augmented Generation Models:** Lewis et al. (2020) introduced retrieval augmentation approach, firstly originating in the field of extractive open-domain question answering (Chen et al., 2017), to sequence-to-sequence models and validated its effectiveness in knowledge-intensive tasks. Contemporary work by Min et al. (2020) applied the retrieval augmentation approach to the task of ambiguous question answering. Izacard and Grave (2021b) proposed Fusion-in-Decoder (FiD), in which each retrieved passage is independently encoded and then jointly input to the decoder. FiD achieves high scalability for the number of passages and effectively aggregates information by jointly using all passages in the decoder.

Recently, the retrieval-augmented language generation approach has received attention, including its incorporation into the language model pretraining (Borgeaud et al., 2022; Izacard et al., 2022; Zhang et al., 2022b), dialogue generation (Komeili et al., 2022), and code generation (Parvez et al.,

2021). The approach has also been shown effective in improving inference of pre-trained language models (e.g., GPT-3 (Brown et al., 2020)) without additional training (Lazaridou et al., 2022; Mallen et al., 2023). For a comprehensive survey on this evolving field, refer to (Yu et al., 2022).

**Effect of Context Characteristics on Retrieval-augmented Models:** Several studies have investigated how context characteristics affect inference of retrieval-augmented generation models. For example, increasing the number of top-ranking passages used as the context has been found to improve performance in question answering (Izacard and Grave, 2021b) and response/prose generation (Zhang et al., 2022b), while a higher proportion of false information in the context degrades performance in question answering (Weller et al., 2022)[16]. Liu et al. (2023) found that the performance of language models on multi-document question answering is influenced by the position of a relevant document.

However, limited knowledge is available regarding the impact of context characteristics on the training of retrieval-augmented geneartion models. Notably, a few existing research suggest that the model performance in question answering im-

---

[16]Du et al. (2022) reported similar results in fact verification, while their focus is not on language generation models.

proves by providing more top-ranking passages during training (Izacard and Grave, 2021b) or by randomly masking top-ranking passages during training (Zhang et al., 2022a), and that the model's memorization behavior is reduced by increasing recall of relevant information in the context during training (Longpre et al., 2021; Chen et al., 2022).

# 6 Conclusion

In this paper, we investigate how context quality and quantity affect the training of FiD models in extractive open-domain question answering tasks. We show that FiD models tend to overfit to the context quality during training, resulting in degraded performance when evaluated in environments with different context qualities. Additionally, our research reveals that the overfitting to context quality is partially explained by different patterns in the model's cross-attention probability. Based on these observations, we propose changing the selectivity of the cross-attention probability to mitigate the effect of overfitting to context quality. The results of this paper suggest a broad spectrum of future work, including more sophisticated adaptation methods and investigations of the effect of other context characteristics on the training of retrieval-augmented generation models.

# 7 Limitations

In this study, we investigated how the quality and quantity of context affect the training of FiD models in extractive open-domain question answering tasks. Our experiments revealed for the first time that context quality significantly impacts FiD models' training and that FiD models tend to overfit to context quality of the training data. The implications of our findings suggest that various context characteristics similarly affect the training of retrieval-augmented generation models, potentially leading to issues such as overfitting.

However, our experiments have several limitations that reduce the generalizability of our findings:

**Task:** Firstly, in this paper, we only focused on the extractive open-domain question answering task, and it is unclear whether similar results can be obtained in other tasks such as dialogue generation, fact verification, code generation, and summarization.

**Model Architecture:** Secondly, our analysis only targeted FiD models, and it is unclear whether different architectures such as RAG (Lewis et al., 2020) and Internet-augmented language models (Lazaridou et al., 2022) produce similar results. Also, it is an interesting direction of future work to conduct similar investigations on non-generative retrieval-augmented models such as FiE (Kedia et al., 2022).

**Model Size:** Thirdly, our experiments are focused on only t5-base, and it is unclear how scaling model size changes the behavior of overfitting to context quality.

**Characteristic of Context:** Lastly, coverage of our analysis is limited to quality and quantity, and further research is required to investigate the effect of other context characteristics. For example, in the field of extractive question answering, it has been shown that models may overfit to answer positions in the contexts (Ko et al., 2020), be misled by adversarially inserted sentence (Jia and Liang, 2017; Jiang and Bansal, 2019), and be susceptible to whether an answer is in the most similar sentence in the context (Sugawara et al., 2018). These findings suggest that those context characteristics may also affect retrieval-augmented generation models.

Other than that, our experiments involved the automatic annotation of relevant and irrelevant passages, which may limit the accuracy of our analysis. Future studies should incorporate human annotation to ensure the high quality of the annotation. Also, passages with similar relevant information can impact models differently due to qualitative factors such as readability and writing style. Nevertheless, as quantitatively evaluating these factors poses challenges, our study did not conduct a fine-grained analysis regarding these aspects.

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

## A Implementation Details

### A.1 Training and Evaluation Data

We trained FiD models with the original train set of each dataset, and we further split the original train set into $\mathcal{D}_{\text{train}}$ for training and $\mathcal{D}_{\text{dev}}$ for evaluating performance during training. To train models in a strictly extractive task environment, we excluded questions for which no retrieved passage contained any of their ground-truth answers. For evaluation, we used the original development set of each dataset as evaluation data $\mathcal{D}_{\text{eval}}$. For a fair comparison, we used the same set of questions with at least 3 (Natural Questions) or 10 (TriviaQA) relevant passages and at least 64 irrelevant passages during training or evaluation. Statistics of the datasets are shown in Table 3.

### A.2 Details of FiD Training and Inference

Our model implementation of FiD, including the loss function for training, is based on the official implementation by Izacard and Grave (2021b)[17]. For both training and inference, we used `transformers` (Ver. 4.23.1) (Wolf et al., 2020). We trained the models with `Seq2SeqTrainer` provided in `transformers`. Hyperparameters for `Seq2SeqTrainer` used in our experiments are listed in Table 4 and other hyperparameters were set to default values. Since we trained models with 32 A100 GPUs, the effective batch size is 64. We used a model checkpoint with highest EM on $\mathcal{D}_{\text{dev}}$ for downstream evaluations.

Since most questions in Natural Questions are annotated with only one ground-truth answer, we used the first ground-truth answer for each question as a target output for model training. On the other

---

[17]https://github.com/facebookresearch/FiD

Table 3: Size of datasets used for training and evaluation.

| Natural Questions | | | TriviaQA | | |
|---|---|---|---|---|---|
| $\mathcal{D}_{\text{train}}$ | $\mathcal{D}_{\text{dev}}$ | $\mathcal{D}_{\text{eval}}$ | $\mathcal{D}_{\text{train}}$ | $\mathcal{D}_{\text{dev}}$ | $\mathcal{D}_{\text{eval}}$ |
| 20728 | 3048 | 2589 | 11414 | 1695 | 1434 |

Table 4: Hyperparameters for `Seq2SeqTrainer`

| parameter | value |
|---|---|
| learning_rate | 0.00005 |
| lr_scheduler_type | constant_with_warmup |
| warmup_steps | 1000 |
| weight_decay | 0.01 |
| max_grad_norm | 1.0 |
| max_steps | 15000 |
| per_device_train_batch_size | 1 |
| gradient_accumulation | 2 |
| eval_steps | 500 |
| save_steps | 500 |
| save_strategy | steps |

hand, since questions in TriviaQA are more exhaustively annotated with paraphrases of ground-truth answers, we randomly sampled one ground-truth answer that appeared in any of the input passages as a target output at every training step.

We tokenized each input passage $\tilde{p}_i$ described in §2.2 and target outputs by the tokenizer of `t5-base`. For both training and inference, to fix the sequence length of each tokenized passage to 256, we conducted truncation for longer passages or padding for shorter passages. We did not truncate target outputs during training and set a maximum length of a predicted answer to 50 during inference.

In experiments where we subsampled passages to control context quality and quantity, to reduce the effect of bias in sampled passages, we sampled different passages at every training step instead of repeatedly using the fixed set of passages sampled before training.

## B Details of Experimental Design

We trained three FiD models with different random seeds for each training environment and conducted evaluations for these models in each evaluation environment. We sampled five different sets of passages for each evaluation environment and computed average model performance on these sets of passages. We independently sampled relevant and irrelevant passages, and thus, the same set of relevant (or irrelevant) passages was sampled regardless of the number of irrelevant (or relevant) passages as long as the number of relevant (or irrel-

Table 5: Performance of FiD models trained in each mixture of environments on TriviaQA. The color intensity indicates the relative performance within the same evaluation mixture of environments (column). Checkmarks indicate which environment included in each mixture.

| $n^+_{eval}(\rightarrow)$ | | | ✓ | | | | ✓ | ✓ | ✓ |
|---|---|---|---|---|---|---|---|---|---|
| $n^+_{train}(\downarrow)$ 5 | | | | ✓ | | ✓ | | ✓ | ✓ |
| 10 | | | | | ✓ | ✓ | ✓ | | ✓ |
| 2 | 5 | 10 | | | | | | | |
| ✓ | | | **79.8** | 93.9 | 98.3 | 96.1 | **89.0** | **86.8** | **90.6** |
| | ✓ | | 75.0 | 93.8 | 98.3 | 96.0 | 86.6 | 84.4 | 89.0 |
| | | ✓ | 61.7 | 90.0 | 97.7 | 93.8 | 79.7 | 75.8 | 83.1 |
| | ✓ | ✓ | 73.4 | 92.4 | 97.7 | 95.1 | 85.6 | 82.9 | 87.9 |
| ✓ | | ✓ | 79.1 | 93.3 | 97.5 | 95.4 | 88.3 | 86.2 | 90.0 |
| ✓ | ✓ | | 79.0 | **94.2** | **98.4** | **96.3** | 88.7 | 86.6 | 90.5 |
| ✓ | ✓ | ✓ | 77.8 | 93.4 | 98.0 | 95.7 | 87.9 | 85.6 | 89.7 |

Table 6: Size of datasets used to train FiD models for the relevant passage annotation

| $\mathcal{D}_{0,train}$ | $\mathcal{D}_{0,dev}$ | $\mathcal{D}_{1,train}$ | $\mathcal{D}_{1,dev}$ |
|---|---|---|---|
| 30612 | 4411 | 30589 | 4373 |
| | (Natural Questions) | | |
| 28696 | 4112 | 28489 | 4115 |
| | (TriviaQA) | | |

evant) passages was the same.

## C  Details of Passage Relevance Annotation by Question Answering Model

We used FiD models as pre-trained question answering models used in the passage relevance annotation described in §2.3, and we trained those models as described in Appendix A except training and development data which we describe below. For each dataset with original train set $\mathcal{D}_{train}$ and development set $\mathcal{D}_{dev}$, we split $\mathcal{D}_{train}$ into four sets: $\mathcal{D}_{0,train}$, $\mathcal{D}_{0,dev}$, $\mathcal{D}_{1,train}$, and $\mathcal{D}_{1,dev}$. Then, we respectively trained a FiD model $\mathcal{M}_0$ or $\mathcal{M}_1$ with a pair of train and development data $(\mathcal{D}_{0,train}, \mathcal{D}_{0,dev})$ or $(\mathcal{D}_{1,train}, \mathcal{D}_{1,dev})$. Finally, we annotated $\mathcal{D}_{0,train}$ and $\mathcal{D}_{0,dev}$ with $\mathcal{M}_1$, and $\mathcal{D}_{1,train}$, $\mathcal{D}_{1,dev}$, and $\mathcal{D}_{dev}$ with $\mathcal{M}_0$. See Table 6 for statistics of the datasets used to train FiD models for the relevant passage annotation.

Our preliminary experiments showed that the behavior of FiD models differs when trained with all passages in the original dataset (**All**), compared to when trained with only those passages containing a ground-truth answer (**Pos**). Thus, we chose a stricter criterion to extract relevant passages. Specifically, we trained $\mathcal{M}_0$ and $\mathcal{M}_1$ and annotated passages in each of **All** and **Pos** setting, and we extracted only those passages annotated as

a relevant passage in the both settings.

## D  Full Experimental Results

Full results of the experiments in §3.1 are shown in Figure 6 and Table 7 for TriviaQA and Figure 7 and Table 8 for Natural Questions.

Full results of the experiments in §3.2 are shown in Figure 8 and Table 9 for TriviaQA and Figure 9 and Table 10 for Natural Questions.

Results of the experiments in §3.3 are shown in Table 5 for TriviaQA.

Results of the experiments in §4.2 are shown in Figure 10.

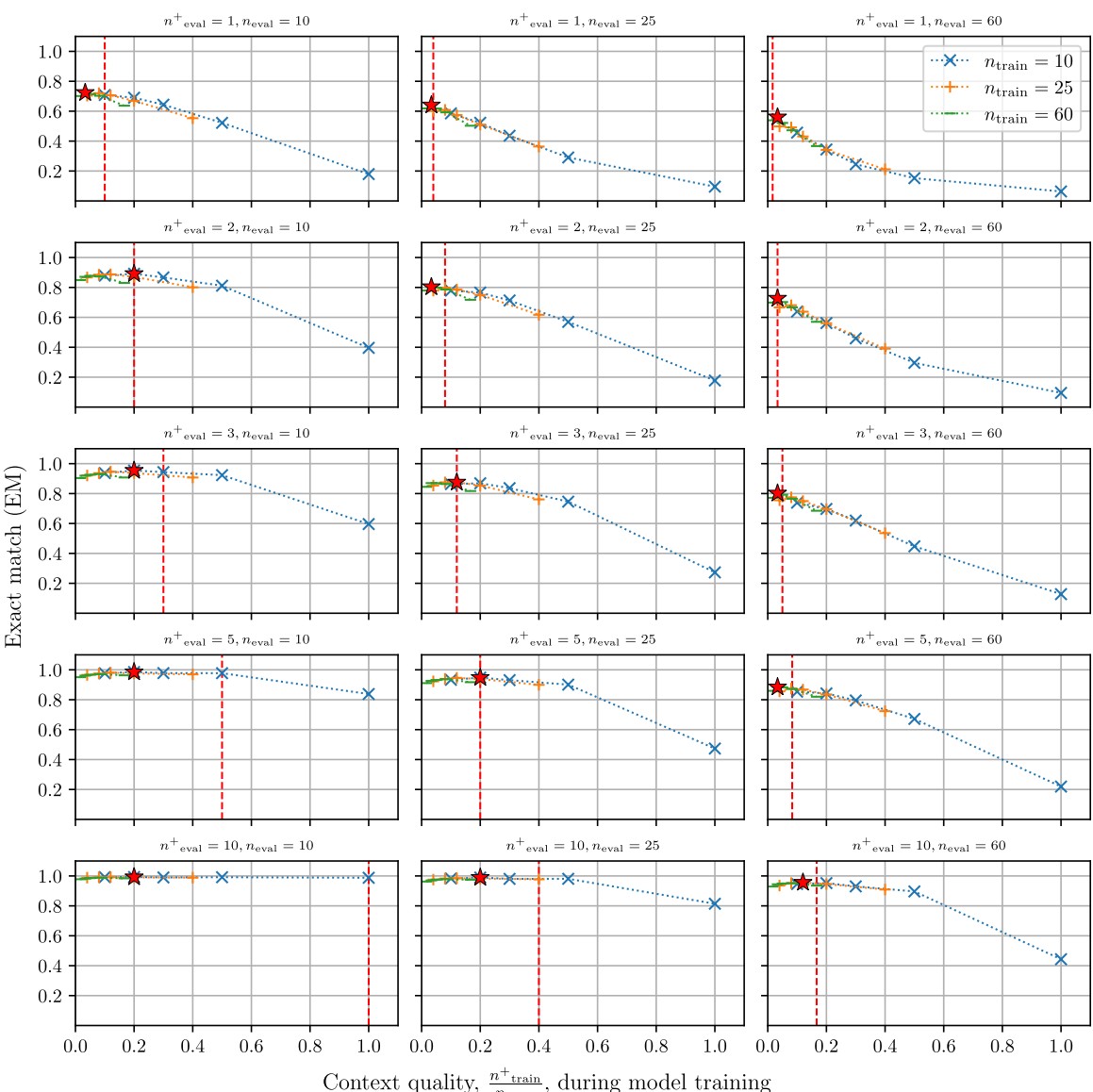

Figure 6: Performance of FiD models on TriviaQA with varying training context quality. Panels represent different evaluation environments with different $(n^+_{\text{eval}}, n_{\text{eval}})$ pairs, and a red dashed line shows corresponding context quality. Red stars represent the best performed models in the corresponding evaluation environments. Dotted lines show models trained on the same context quantity $n_{\text{train}}$.

Table 7: Exact match (EM) [%] of FiD models trained in environment $(n^+_{\text{train}}, n_{\text{train}})$ in various evaluation environment $(n^+_{\text{eval}}, n_{\text{eval}})$ in TriviaQA. The standard deviations for each reported value is denoted with lower subscripts.

| $n^+_{\text{eval}}$ | $n^+_{\text{train}}$ | $n_{\text{eval}}$ = 10 | | | 25 | | | 60 | | |
| | $n_{\text{train}}$ = | 10 | 25 | 60 | 10 | 25 | 60 | 10 | 25 | 60 |
|---|---|---|---|---|---|---|---|---|---|---|
| 1 | 1 | $70.9_{3.0}$ | $70.4_{1.5}$ | $70.1_{1.9}$ | $58.5_{3.1}$ | $60.1_{1.6}$ | $61.9_{3.1}$ | $45.7_{3.5}$ | $49.8_{1.8}$ | $53.9_{3.4}$ |
| | 2 | $69.0_{1.2}$ | $72.1_{2.7}$ | $72.4_{1.1}$ | $52.3_{1.9}$ | $60.8_{3.4}$ | $64.1_{2.0}$ | $34.4_{1.9}$ | $49.2_{4.0}$ | $56.1_{2.6}$ |
| | 3 | $64.3_{0.5}$ | $70.7_{0.8}$ | $71.4_{1.1}$ | $43.6_{0.5}$ | $57.4_{1.6}$ | $61.8_{1.7}$ | $24.5_{0.2}$ | $43.1_{2.3}$ | $52.2_{2.2}$ |
| | 5 | $52.2_{0.8}$ | $66.9_{0.9}$ | $70.3_{2.0}$ | $29.1_{1.3}$ | $51.0_{1.7}$ | $59.2_{2.9}$ | $15.2_{0.5}$ | $34.1_{2.4}$ | $47.2_{3.3}$ |
| | 10 | $18.0_{0.6}$ | $55.3_{1.2}$ | $63.7_{1.8}$ | $9.5_{0.2}$ | $36.4_{0.5}$ | $50.3_{1.4}$ | $6.4_{0.2}$ | $21.2_{0.4}$ | $36.7_{0.6}$ |
| 2 | 1 | $87.9_{1.4}$ | $86.7_{0.6}$ | $84.9_{0.2}$ | $77.9_{2.2}$ | $77.9_{1.3}$ | $77.9_{1.7}$ | $63.9_{3.3}$ | $66.6_{1.6}$ | $69.9_{3.1}$ |
| | 2 | $89.1_{0.5}$ | $88.5_{1.0}$ | $87.1_{0.4}$ | $76.6_{1.0}$ | $79.8_{2.0}$ | $80.4_{1.1}$ | $56.2_{2.0}$ | $68.1_{3.5}$ | $72.7_{2.1}$ |
| | 3 | $86.7_{0.3}$ | $88.7_{0.4}$ | $87.8_{0.4}$ | $71.3_{0.2}$ | $78.6_{1.0}$ | $79.8_{1.1}$ | $45.9_{0.6}$ | $63.8_{2.3}$ | $70.2_{1.5}$ |
| | 5 | $81.2_{0.5}$ | $87.0_{0.2}$ | $87.3_{0.7}$ | $56.9_{1.2}$ | $75.0_{0.5}$ | $78.6_{1.7}$ | $29.6_{1.4}$ | $56.1_{2.2}$ | $66.7_{2.8}$ |
| | 10 | $39.6_{0.6}$ | $80.1_{0.7}$ | $83.0_{1.1}$ | $17.8_{0.6}$ | $61.7_{0.7}$ | $71.7_{1.4}$ | $9.5_{0.3}$ | $39.0_{0.9}$ | $57.0_{1.3}$ |
| 3 | 1 | $93.7_{0.7}$ | $92.1_{0.1}$ | $90.4_{0.4}$ | $86.3_{1.6}$ | $85.4_{0.9}$ | $84.5_{0.8}$ | $73.9_{2.9}$ | $75.5_{1.4}$ | $77.2_{2.2}$ |
| | 2 | $95.4_{0.1}$ | $93.6_{0.2}$ | $92.0_{0.1}$ | $86.9_{0.8}$ | $87.5_{1.3}$ | $87.0_{0.6}$ | $69.7_{1.9}$ | $77.5_{3.0}$ | $80.2_{1.4}$ |
| | 3 | $94.3_{0.4}$ | $94.7_{0.2}$ | $92.6_{0.4}$ | $83.6_{0.5}$ | $87.6_{0.4}$ | $87.0_{0.5}$ | $61.9_{0.9}$ | $75.0_{1.7}$ | $79.2_{1.3}$ |
| | 5 | $92.3_{0.4}$ | $93.7_{0.5}$ | $93.1_{0.1}$ | $74.6_{0.7}$ | $85.4_{0.1}$ | $86.4_{0.9}$ | $44.7_{1.1}$ | $69.6_{1.5}$ | $76.5_{2.1}$ |
| | 10 | $59.6_{0.4}$ | $90.8_{0.2}$ | $90.8_{0.7}$ | $27.3_{0.4}$ | $76.1_{0.3}$ | $81.7_{0.8}$ | $12.8_{0.1}$ | $53.6_{0.7}$ | $68.5_{0.8}$ |
| 5 | 1 | $97.7_{0.3}$ | $96.6_{0.1}$ | $95.1_{1.0}$ | $93.3_{0.8}$ | $92.3_{0.4}$ | $91.1_{0.1}$ | $85.4_{1.8}$ | $85.6_{0.7}$ | $85.9_{1.1}$ |
| | 2 | $98.5_{0.1}$ | $97.5_{0.4}$ | $96.1_{0.4}$ | $94.8_{0.4}$ | $93.9_{0.3}$ | $92.6_{0.0}$ | $84.2_{0.9}$ | $87.5_{1.4}$ | $88.5_{0.4}$ |
| | 3 | $97.8_{0.1}$ | $98.0_{0.1}$ | $96.6_{0.6}$ | $93.1_{0.1}$ | $94.6_{0.4}$ | $93.2_{0.3}$ | $79.5_{0.6}$ | $86.8_{0.9}$ | $88.3_{0.5}$ |
| | 5 | $97.7_{0.2}$ | $97.6_{0.5}$ | $97.1_{0.2}$ | $90.1_{0.6}$ | $93.8_{0.6}$ | $93.8_{0.1}$ | $67.2_{0.8}$ | $83.5_{0.3}$ | $87.2_{0.8}$ |
| | 10 | $83.8_{0.2}$ | $97.0_{0.3}$ | $96.3_{0.2}$ | $47.4_{0.6}$ | $90.0_{0.2}$ | $91.7_{0.2}$ | $21.9_{0.3}$ | $72.4_{0.2}$ | $82.0_{0.9}$ |
| 10 | 1 | $99.2_{0.2}$ | $98.9_{0.1}$ | $97.8_{0.9}$ | $98.3_{0.1}$ | $97.5_{0.1}$ | $96.3_{0.5}$ | $94.6_{0.6}$ | $93.6_{0.1}$ | $92.9_{0.3}$ |
| | 2 | $99.3_{0.1}$ | $99.0_{0.1}$ | $98.2_{0.1}$ | $98.9_{0.0}$ | $98.3_{0.4}$ | $97.0_{0.3}$ | $95.2_{0.3}$ | $95.2_{0.0}$ | $94.4_{0.2}$ |
| | 3 | $99.0_{0.2}$ | $99.2_{0.1}$ | $98.6_{0.4}$ | $98.0_{0.2}$ | $98.5_{0.1}$ | $97.5_{0.4}$ | $93.0_{0.2}$ | $95.6_{0.3}$ | $94.9_{0.3}$ |
| | 5 | $99.2_{0.2}$ | $99.0_{0.3}$ | $98.9_{0.3}$ | $98.1_{0.2}$ | $98.3_{0.6}$ | $97.9_{0.3}$ | $89.6_{0.6}$ | $94.6_{0.7}$ | $95.2_{0.6}$ |
| | 10 | $98.8_{0.1}$ | $98.9_{0.2}$ | $98.4_{0.4}$ | $81.4_{0.4}$ | $97.7_{0.4}$ | $97.5_{0.2}$ | $44.3_{0.3}$ | $90.9_{0.4}$ | $93.5_{0.5}$ |

Table 8: Exact match (EM) [%] of FiD models trained in environment $(n^+_{\text{train}}, n_{\text{train}})$ in various evaluation environment $(n^+_{\text{eval}}, n_{\text{eval}})$ in Natural Questions. The standard deviations for each reported value is denoted with lower subscripts.

| $n^+_{\text{eval}}$ | $n^+_{\text{train}}$ | $n_{\text{eval}}$ = 10 | | | 25 | | | 60 | | |
| | $n_{\text{train}}$ = | 10 | 25 | 60 | 10 | 25 | 60 | 10 | 25 | 60 |
|---|---|---|---|---|---|---|---|---|---|---|
| 1 | 1 | $69.8_{0.7}$ | $70.3_{0.3}$ | $65.8_{0.3}$ | $58.1_{0.6}$ | $62.4_{0.7}$ | $61.0_{0.1}$ | $45.5_{0.8}$ | $54.1_{1.0}$ | $55.9_{0.4}$ |
| | 2 | $65.0_{0.5}$ | $69.6_{0.2}$ | $68.1_{0.5}$ | $45.4_{1.2}$ | $58.6_{0.5}$ | $60.7_{0.8}$ | $24.1_{1.5}$ | $45.7_{0.8}$ | $53.2_{1.2}$ |
| | 3 | $53.9_{1.2}$ | $65.8_{0.2}$ | $67.8_{0.4}$ | $29.0_{0.4}$ | $50.7_{0.8}$ | $58.7_{0.2}$ | $10.6_{0.2}$ | $33.3_{2.1}$ | $49.2_{0.3}$ |
| 2 | 1 | $82.9_{0.4}$ | $80.3_{0.3}$ | $74.0_{1.4}$ | $73.7_{0.2}$ | $74.1_{0.2}$ | $70.8_{0.8}$ | $61.9_{0.5}$ | $67.0_{0.6}$ | $66.5_{0.3}$ |
| | 2 | $85.5_{0.1}$ | $83.9_{0.3}$ | $79.9_{0.1}$ | $71.4_{0.5}$ | $75.7_{0.1}$ | $74.5_{0.4}$ | $48.2_{1.4}$ | $64.3_{0.6}$ | $67.8_{0.7}$ |
| | 3 | $83.4_{0.1}$ | $84.2_{0.3}$ | $81.9_{0.7}$ | $60.9_{0.6}$ | $73.9_{0.3}$ | $75.5_{0.8}$ | $29.8_{0.1}$ | $57.6_{1.3}$ | $66.6_{0.7}$ |
| 3 | 1 | $88.6_{0.8}$ | $84.4_{0.5}$ | $77.4_{1.6}$ | $80.8_{0.5}$ | $79.2_{0.2}$ | $74.4_{1.0}$ | $70.3_{0.1}$ | $72.9_{0.4}$ | $70.8_{0.9}$ |
| | 2 | $92.2_{0.2}$ | $89.5_{0.6}$ | $84.6_{0.8}$ | $82.4_{0.5}$ | $83.2_{0.5}$ | $79.6_{0.1}$ | $63.5_{1.0}$ | $73.6_{0.5}$ | $74.4_{0.5}$ |
| | 3 | $93.0_{0.1}$ | $91.2_{0.5}$ | $87.6_{0.6}$ | $78.8_{0.3}$ | $83.7_{0.4}$ | $82.1_{0.8}$ | $48.0_{0.2}$ | $70.5_{0.5}$ | $74.8_{0.6}$ |

Table 9: Exact match (EM) [%] of FiD models trained in environment $(n^+_{\text{train}}, k_{\text{train}})$ in various evaluation environment $(n^+_{\text{eval}}, k_{\text{eval}})$ in TriviaQA. The standard deviations for each reported value is denoted with lower subscripts.

| $n^+_{\text{eval}}$ | $k_{\text{eval}}$ / $n^+_{\text{train}}$ / $k_{\text{train}}$ | 1 | | | 5 | | | 20 | | |
|---|---|---|---|---|---|---|---|---|---|---|
| | | 1 | 5 | 20 | 1 | 5 | 20 | 1 | 5 | 20 |
| 1 | 1 | $90.8_{0.2}$ | $90.1_{1.4}$ | $88.6_{0.9}$ | $77.4_{0.3}$ | $76.9_{2.3}$ | $77.5_{2.5}$ | $53.5_{0.3}$ | $57.6_{3.6}$ | $63.4_{3.5}$ |
| | 2 | $89.0_{0.2}$ | $89.4_{0.6}$ | $88.3_{0.2}$ | $69.0_{0.1}$ | $77.6_{0.4}$ | $77.1_{0.4}$ | $32.2_{1.1}$ | $58.7_{1.0}$ | $63.4_{1.1}$ |
| | 3 | $88.0_{0.4}$ | $89.4_{0.3}$ | $87.8_{0.8}$ | $68.8_{1.8}$ | $77.1_{0.8}$ | $77.1_{1.5}$ | $34.8_{3.5}$ | $58.3_{1.1}$ | $63.7_{2.9}$ |
| | 5 | $85.8_{0.3}$ | $87.6_{0.3}$ | | $65.4_{0.8}$ | $74.0_{0.5}$ | | $33.3_{1.3}$ | $54.7_{1.8}$ | |
| | 8 | $84.0_{1.1}$ | $86.1_{0.5}$ | | $64.0_{1.6}$ | $72.8_{0.7}$ | | $35.3_{1.9}$ | $53.9_{2.0}$ | |
| | 10 | $83.6_{0.6}$ | $84.3_{1.2}$ | | $63.9_{0.6}$ | $70.7_{1.8}$ | | $35.3_{0.7}$ | $52.7_{1.2}$ | |
| 2 | 1 | $95.4_{0.2}$ | $95.1_{0.4}$ | $93.7_{0.6}$ | $85.1_{0.5}$ | $85.8_{1.2}$ | $85.7_{1.4}$ | $61.0_{0.3}$ | $66.4_{3.4}$ | $73.0_{3.2}$ |
| | 2 | $95.6_{0.1}$ | $95.2_{0.5}$ | $93.9_{0.2}$ | $81.2_{0.2}$ | $87.0_{0.6}$ | $85.9_{0.4}$ | $38.0_{1.2}$ | $68.4_{1.0}$ | $73.9_{1.4}$ |
| | 3 | $95.0_{0.1}$ | $94.6_{0.5}$ | $93.5_{0.1}$ | $81.1_{1.1}$ | $86.6_{0.7}$ | $86.3_{0.7}$ | $41.5_{3.4}$ | $68.4_{0.7}$ | $74.8_{2.5}$ |
| | 5 | $93.6_{0.1}$ | $93.9_{0.3}$ | | $77.9_{0.4}$ | $84.9_{0.6}$ | | $40.2_{1.4}$ | $65.6_{2.1}$ | |
| | 8 | $92.0_{0.7}$ | $92.8_{0.3}$ | | $75.4_{1.5}$ | $83.1_{0.3}$ | | $42.1_{1.9}$ | $64.6_{1.9}$ | |
| | 10 | $91.5_{0.2}$ | $91.6_{0.8}$ | | $74.6_{0.2}$ | $81.3_{1.3}$ | | $41.8_{0.4}$ | $63.3_{1.5}$ | |
| 3 | 1 | $96.9_{0.1}$ | $96.7_{0.3}$ | $95.0_{0.3}$ | $88.2_{0.2}$ | $89.4_{0.9}$ | $88.9_{0.9}$ | $63.5_{0.6}$ | $69.8_{3.2}$ | $76.5_{3.5}$ |
| | 2 | $97.3_{0.3}$ | $96.6_{0.5}$ | $95.3_{0.2}$ | $84.9_{0.2}$ | $90.6_{0.7}$ | $89.5_{0.3}$ | $39.9_{1.2}$ | $71.3_{0.9}$ | $77.7_{1.2}$ |
| | 3 | $97.2_{0.1}$ | $96.2_{0.3}$ | $95.2_{0.1}$ | $85.7_{1.3}$ | $90.0_{0.5}$ | $89.6_{0.6}$ | $43.9_{3.5}$ | $71.3_{0.4}$ | $78.5_{2.1}$ |
| | 5 | $96.2_{0.2}$ | $95.9_{0.1}$ | | $82.9_{0.4}$ | $89.2_{0.2}$ | | $43.0_{1.0}$ | $69.3_{1.6}$ | |
| | 8 | $95.1_{0.1}$ | $95.0_{0.4}$ | | $80.9_{1.0}$ | $87.1_{0.3}$ | | $45.4_{2.4}$ | $68.9_{2.4}$ | |
| | 10 | $94.4_{0.1}$ | $93.7_{0.5}$ | | $80.2_{0.3}$ | $86.0_{0.7}$ | | $45.1_{0.3}$ | $67.6_{1.0}$ | |
| 5 | 1 | $97.7_{0.2}$ | $98.0_{0.0}$ | $96.8_{0.4}$ | $90.0_{0.3}$ | $92.0_{0.4}$ | $91.6_{0.5}$ | | | |
| | 2 | $98.0_{0.1}$ | $97.9_{0.3}$ | $96.9_{0.5}$ | $87.8_{0.3}$ | $93.2_{0.4}$ | $92.3_{0.1}$ | | | |
| | 3 | $98.1_{0.2}$ | $97.6_{0.3}$ | $96.9_{0.4}$ | $88.4_{1.1}$ | $92.3_{0.2}$ | $92.4_{0.5}$ | | | |
| | 5 | $97.7_{0.2}$ | $97.7_{0.1}$ | | $86.5_{0.3}$ | $92.5_{0.3}$ | | | | |
| | 8 | $97.1_{0.2}$ | $97.1_{0.2}$ | | $84.9_{1.1}$ | $90.8_{0.3}$ | | | | |
| | 10 | $96.8_{0.1}$ | $96.3_{0.2}$ | | $84.1_{0.4}$ | $90.3_{0.3}$ | | | | |
| 8 | 1 | $98.1_{0.0}$ | $98.7_{0.1}$ | $97.4_{0.6}$ | $91.3_{0.2}$ | $93.7_{0.2}$ | $93.3_{0.2}$ | | | |
| | 2 | $98.4_{0.1}$ | $98.5_{0.3}$ | $97.8_{0.3}$ | $89.1_{0.4}$ | $94.5_{0.3}$ | $94.2_{0.1}$ | | | |
| | 3 | $98.6_{0.3}$ | $98.2_{0.2}$ | $97.6_{0.3}$ | $90.2_{1.0}$ | $93.9_{0.4}$ | $94.5_{0.2}$ | | | |
| | 5 | $98.4_{0.2}$ | $98.4_{0.3}$ | | $88.8_{0.2}$ | $94.2_{0.4}$ | | | | |
| | 8 | $97.9_{0.3}$ | $98.0_{0.3}$ | | $87.7_{0.7}$ | $93.3_{0.4}$ | | | | |
| | 10 | $97.9_{0.0}$ | $97.4_{0.3}$ | | $87.0_{0.2}$ | $92.6_{0.4}$ | | | | |
| 10 | 1 | $98.5_{0.1}$ | $99.0_{0.1}$ | $98.0_{0.6}$ | $91.9_{0.4}$ | $94.4_{0.1}$ | $94.1_{0.2}$ | | | |
| | 2 | $98.6_{0.2}$ | $98.7_{0.2}$ | $98.1_{0.2}$ | $90.2_{0.2}$ | $94.9_{0.3}$ | $95.0_{0.5}$ | | | |
| | 3 | $98.7_{0.1}$ | $98.5_{0.3}$ | $98.0_{0.3}$ | $90.9_{1.1}$ | $94.6_{0.4}$ | $95.1_{0.3}$ | | | |
| | 5 | $98.7_{0.1}$ | $98.7_{0.1}$ | | $89.6_{0.6}$ | $95.0_{0.5}$ | | | | |
| | 8 | $98.3_{0.2}$ | $98.3_{0.3}$ | | $88.7_{0.7}$ | $94.0_{0.3}$ | | | | |
| | 10 | $98.1_{0.1}$ | $97.9_{0.3}$ | | $88.4_{0.2}$ | $93.5_{0.5}$ | | | | |

Table 10: Exact match (EM) [%] of FiD models trained in environment $(n^+_{\text{train}}, k_{\text{train}})$ in various evaluation environment $(n^+_{\text{eval}}, k_{\text{eval}})$ in Natural Questions. The standard deviations for each reported value is denoted with lower subscripts.

| $n^+_{\text{eval}}$ | $k_{\text{eval}}$ / $n^+_{\text{train}}$ / $k_{\text{train}}$ | 1 | | | 5 | | | 20 | | |
|---|---|---|---|---|---|---|---|---|---|---|
| | | 1 | 5 | 20 | 1 | 5 | 20 | 1 | 5 | 20 |
| 1 | 1 | $87.9_{0.5}$ | $88.2_{0.3}$ | $84.3_{0.5}$ | $72.4_{0.7}$ | $76.0_{0.3}$ | $74.7_{0.6}$ | $48.2_{0.6}$ | $58.7_{0.9}$ | $62.9_{1.4}$ |
| | 2 | $85.1_{0.5}$ | $87.6_{0.1}$ | $83.2_{0.8}$ | $59.5_{1.2}$ | $73.9_{0.4}$ | $73.9_{0.7}$ | $20.7_{1.3}$ | $53.2_{0.6}$ | $62.4_{1.0}$ |
| | 3 | $82.4_{0.5}$ | $85.8_{0.3}$ | $82.3_{0.7}$ | $56.0_{1.1}$ | $70.9_{0.9}$ | $72.8_{0.6}$ | $20.4_{1.2}$ | $47.5_{1.0}$ | $60.9_{0.6}$ |
| 2 | 1 | $91.8_{0.5}$ | $91.1_{0.4}$ | $87.6_{0.6}$ | $80.0_{0.8}$ | $82.3_{0.4}$ | $80.2_{0.5}$ | $54.9_{1.1}$ | $65.7_{0.3}$ | $69.3_{0.9}$ |
| | 2 | $93.8_{0.0}$ | $92.4_{0.2}$ | $87.7_{1.0}$ | $75.2_{1.0}$ | $83.5_{0.1}$ | $80.6_{0.2}$ | $25.6_{1.5}$ | $62.4_{0.4}$ | $70.7_{0.6}$ |
| | 3 | $93.1_{0.2}$ | $92.1_{0.2}$ | $87.4_{0.9}$ | $73.9_{0.6}$ | $82.7_{0.6}$ | $80.5_{0.6}$ | $26.3_{1.4}$ | $58.9_{0.8}$ | $70.7_{0.4}$ |
| 3 | 1 | $93.3_{0.3}$ | $92.6_{0.3}$ | $89.2_{0.3}$ | $82.6_{0.8}$ | $84.3_{0.4}$ | $81.7_{0.4}$ | $57.3_{0.9}$ | $68.2_{0.0}$ | $71.5_{0.8}$ |
| | 2 | $96.4_{0.2}$ | $94.6_{0.1}$ | $89.3_{0.8}$ | $80.0_{0.5}$ | $86.7_{0.2}$ | $83.4_{0.4}$ | $27.5_{1.4}$ | $65.7_{0.4}$ | $73.8_{0.4}$ |
| | 3 | $96.2_{0.3}$ | $94.8_{0.2}$ | $89.5_{0.7}$ | $79.8_{0.1}$ | $87.1_{0.3}$ | $83.3_{0.4}$ | $28.4_{0.9}$ | $63.5_{1.0}$ | $74.4_{0.4}$ |

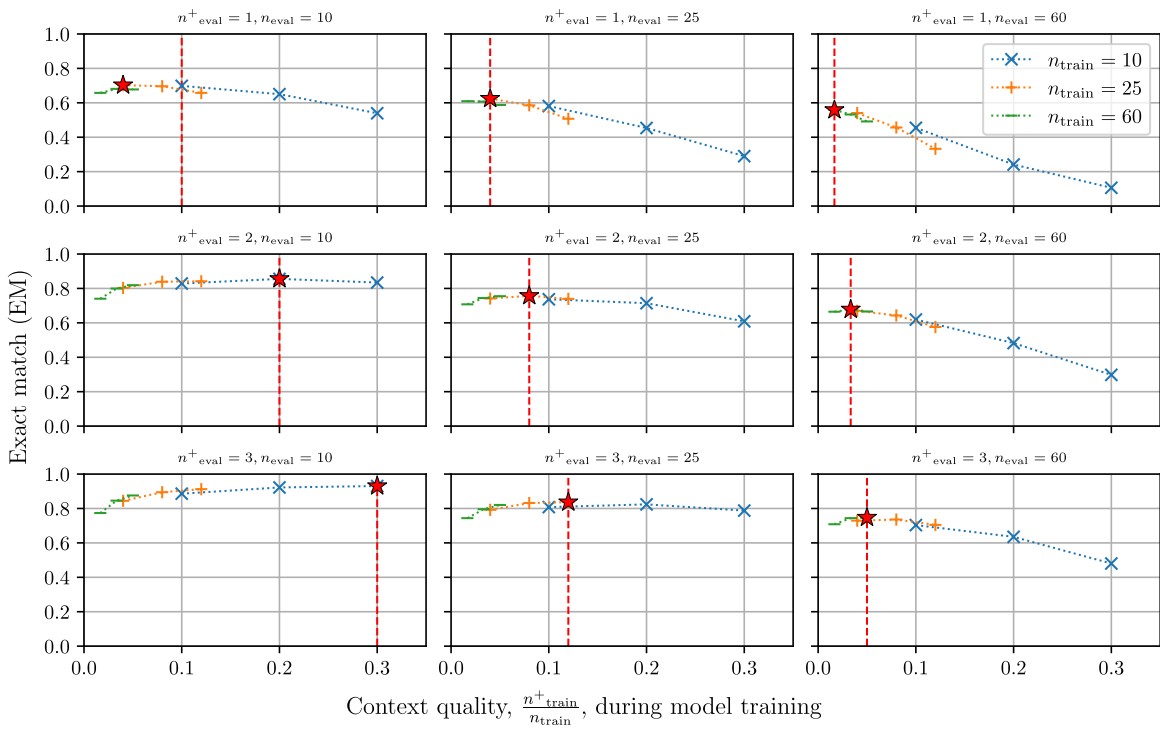

Figure 7: Performance of FiD models on Natural Questions with varying training context quality. Panels represent different evaluation environments with different $(n^+_{\text{eval}}, n_{\text{eval}})$ pairs, and a red dashed line shows corresponding context quality. Red stars represent the best performed models in the corresponding evaluation environments. Dotted lines show models trained on the same context quantity $n_{\text{train}}$.

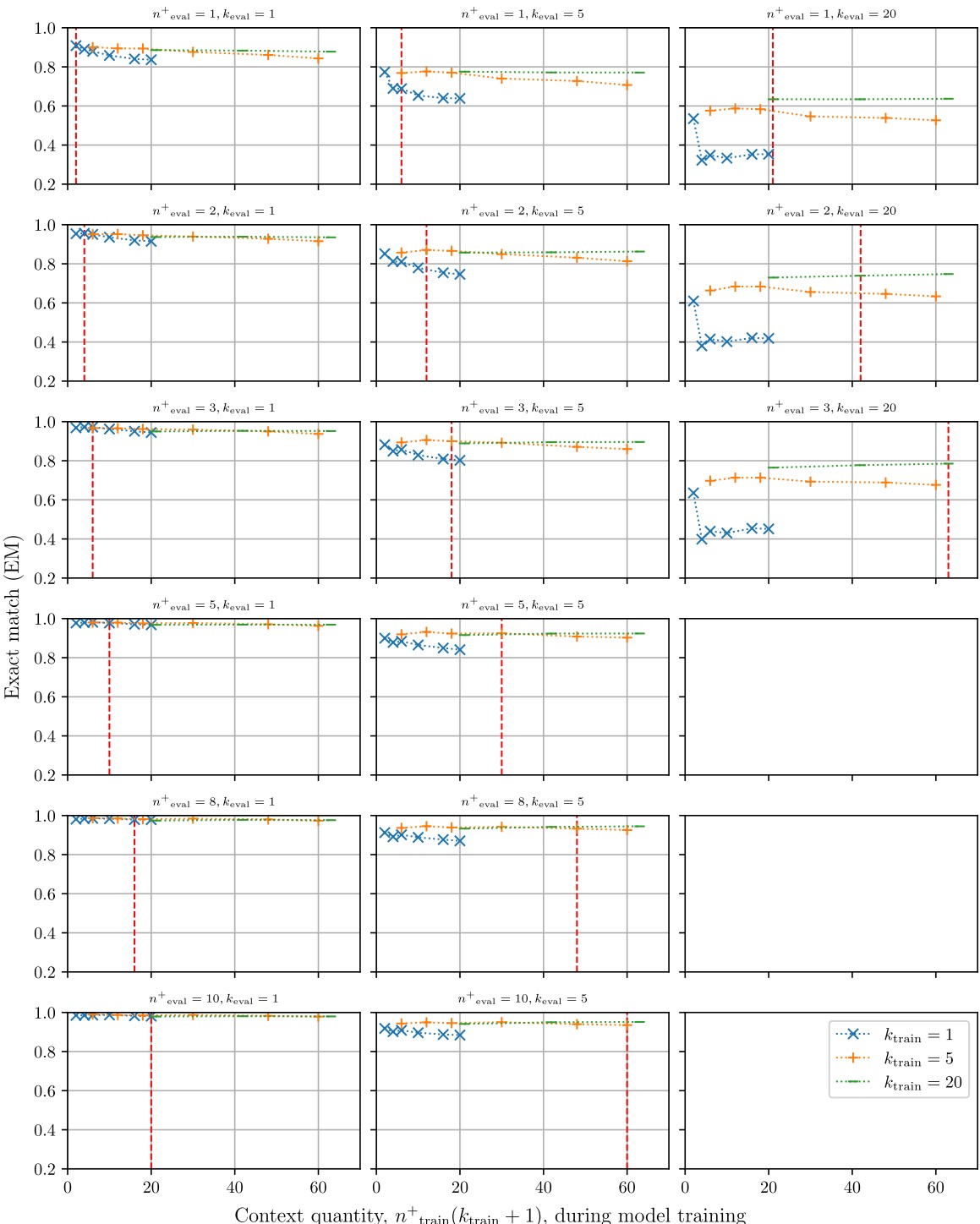

Figure 8: Performance of FiD models on TriviaQA with varying training context quantity. Panels represent different evaluation environments with different $(n^+_{\text{eval}}, k_{\text{eval}})$ pairs, and a red dashed line shows corresponding context quantity. Red stars represent the best performed models in the corresponding evaluation environments. Dotted lines show models trained on the same context quality $\frac{1}{1+k_{\text{train}}}$.

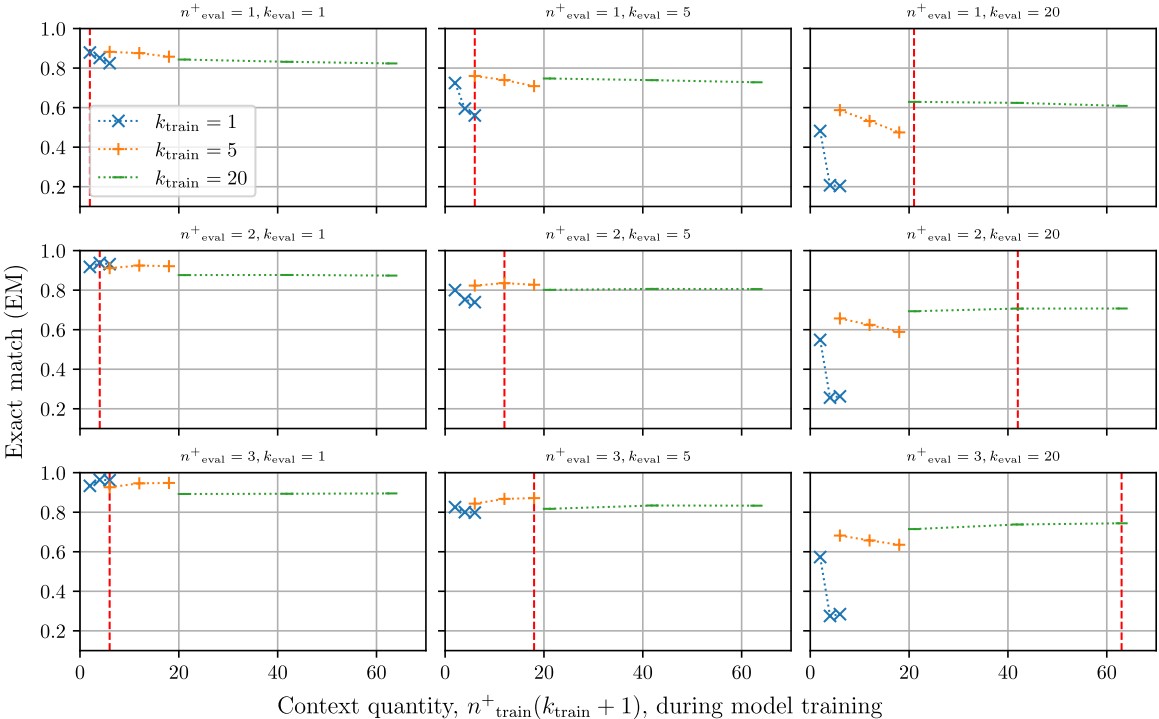

Figure 9: Performance of FiD models on Natural Questions with varying training context quantity. Panels represent different evaluation environments with different $(n^+_{\text{eval}}, k_{\text{eval}})$ pairs, and a red dashed line shows corresponding context quantity. Red stars represent the best performed models in the corresponding evaluation environments. Dotted lines show models trained on the same context quality $\frac{1}{1+k_{\text{train}}}$.

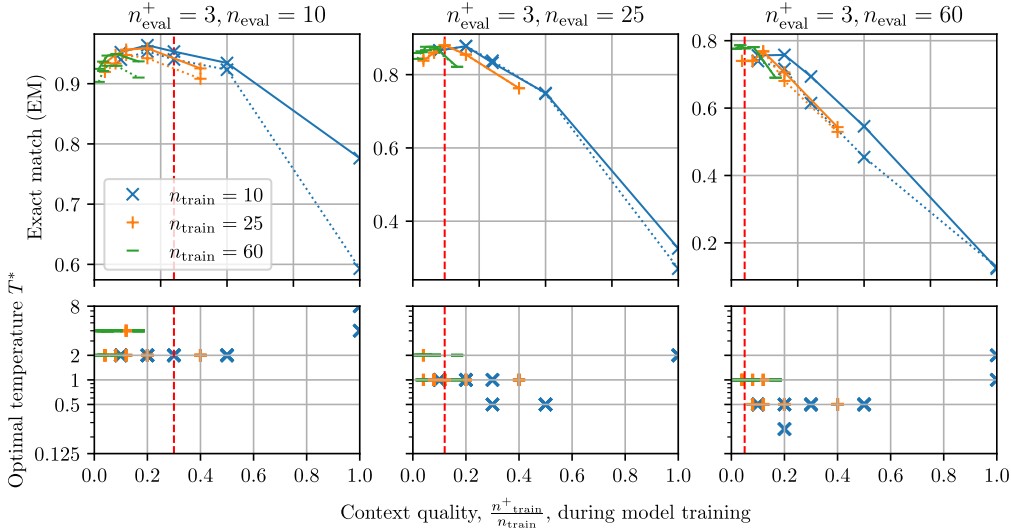

Figure 10: **Top panels:** Performance of FiD models on TriviaQA **with** adaptation by the proposed method (solid lines) and **without** adaptation (dotted lines). **Bottom panels:** Optimal temperature parameter $T^*$ selected for each model. Multiple $T^*$ were selected for some context qualities, i.e., training environments, because we selected $T^*$ for each of the three models trained with different random seeds for each training environment.

Panels represent different evaluation environments with different $(n^+_{\text{eval}}, n_{\text{eval}})$ pairs, and a red dashed line shows corresponding context quality.