# OpenReview forum: "Context Quality Matters in Training Fusion-in-Decoder for Extractive Open-Domain Question Answering"
_EMNLP/2023/Conference — EMNLP 2023 Findings_

### Official Review · Reviewer_wXc7 · 2023-08-04

**Soundness:** 3

**Excitement:**

4: Strong: This paper deepens the understanding of some phenomenon or lowers the barriers to an existing research direction.

**Paper Topic And Main Contributions:**

The paper investigates how the quality and quantity of context affects the performance of Fushion-in-Decoder for retrieval augmented generation. The authors proposes a method for measuring the quality of the context based on the ratio of the relevant of passage to the non-relevant ones. A number of experiments are designed where the main findings are: 1) the training of Fid is overfit to the quality of context; 2) the overfitting is partially due to that training with low quality context makes the model pays more attention to the relevant passages. The authors then propose a simple method to adapt the cross-attention probability during inference, which is shown to outperform FiD.

**Questions For The Authors:**

1. Can you provide intuition behind different values of r in 3.4.2. For r=0 or close to 0, the intervention makes the model
2. Line 251 to 253, different question may have different number of relevant passages, what would you do for questions of which the number of relevant passages is smaller than a specific setting (e.g. n+=3)?
3. Figure 2, 5, red dashed lines show the corresponding context quality in evaluation? If so, please state it clearly.

**Reasons To Accept:**

1. The research question and Experimental designs are interesting
2. The findings of the paper is meaningful for future research
3. The proposed method is simple and easy to be adopted.

**Reasons To Reject:**

1. The findings and conclusion might be limited to extractive open-domain QA task (in Natural Question, and TriviaQA) that is considered in this paper. More experiments should be conducted to support the findings, or the conclusion should be stated more precisely to avoid future misleading.
2. Some results are confusing and needs further explanation.  For example, Table 2 shows that the model trained with low-quality training environment attend more to the relevant passages (during inference?). If keep the evaluation context is of high quality, training with low-quality would be better since we would like the model to attend more to the relevant passages. However, Figure 1 (left) shows that training with high quality would be better.


**Reproducibility:**

4: Could mostly reproduce the results, but there may be some variation because of sample variance or minor variations in their interpretation of the protocol or method.

**Reviewer Confidence:**

4: Quite sure. I tried to check the important points carefully. It's unlikely, though conceivable, that I missed something that should affect my ratings.

**Typos Grammar Style And Presentation Improvements:**

Line 403: Interventioal Experiment > Intervention Experiment
Figure 4: Evaluation environment (n+, n) or Evaluation environment (n+, n-)

---

> ### Author Rebuttal · Authors · 2023-08-28
>
> **Comments for Reasons To Reject section**
>
> > 1. The findings and conclusion might be limited to extractive open-domain QA task (in Natural Question, and TriviaQA) that is considered in this paper. More experiments should be conducted to support the findings, or the conclusion should be stated more precisely to avoid future misleading.
>
> We agree that our findings and conclusions should be more precisely stated as focused on extractive open-domain QA tasks. We will clarify this in the revised version.
>
> > 2. Some results are confusing and needs further explanation. For example, Table 2 shows that the model trained with low-quality training environment attend more to the relevant passages (during inference?). If keep the evaluation context is of high quality, training with low-quality would be better since we would like the model to attend more to the relevant passages. However, Figure 1 (left) shows that training with high quality would be better.
>
> We, too, initially expected that training in a low-quality environment and learning to attend more to relevant passages would always be desirable, and were surprised by the results. While further investigation is needed to determine the cause, we hypothesize that one reason might be that training in a low-quality environment leads to excessive selectivity that causes the model to focus on only a subset of relevant passages. Such excessive selectivity makes the model overlook necessary information and, as a result, fail to correctly answer the questions. In fact, looking at the results in Fig. 3, we can see more disparities among the attention probabilities to relevant passages for the models trained in low-quality environments. It may be the case that, when evaluated in a high-quality environment (i.e., where the majority of passages are relevant to the question), it is more optimal for the model to examine all passages more uniformly without being overly selective. We will include this discussion in the revised version.
>
> **Comments for Questions For The Authors section**
>
> > 1. Can you provide intuition behind different values of r in 3.4.2. For r=0 or close to 0, the intervention makes the model
>
> When r=0, the model completely ignores irrelevant passages, whereas when r=1, the model attends uniformly to all passages.
>
> > 2. Line 251 to 253, different question may have different number of relevant passages, what would you do for questions of which the number of relevant passages is smaller than a specific setting (e.g. n+=3)?
>
> In all experimental settings, we have carefully selected questions that have a sufficient number of relevant passages to conduct our experiments. For example, in our experiments with Natural Questions, the maximum value for n+ is 3, and therefore, we only use questions with three or more relevant passages as training and evaluation data in the experiments (Lines 881-884).
>
> > 3. Figure 2, 5, red dashed lines show the corresponding context quality in evaluation? If so, please state it clearly.
>
> You are correct. The red dashed lines in Figures 2 and 5 represent the corresponding context quality during evaluation. We appreciate your suggestion and will revise the captions to clearly state this point.

---

### Official Review · Reviewer_t9yE · 2023-08-04

**Soundness:** 4

**Excitement:**

3: Ambivalent: It has merits (e.g., it reports state-of-the-art results, the idea is nice), but there are key weaknesses (e.g., it describes incremental work), and it can significantly benefit from another round of revision. However, I won't object to accepting it if my co-reviewers champion it.

**Paper Topic And Main Contributions:**

This paper examines how the retrieval-augmented generation model is affected by the quantity (number of evidence documents) and quality (proportion of question-relevant documents among evidence documents) of context. Through experiments, the authors revealed that context quality is more influential than quantity, and that the discrepancy of context quality during training and inference causes performance degradation. Based on these observations and analyzes, the authors proposed a method to mitigate performance degradation caused by context quality discrepancy in train-inference by adjusting the cross-attention probability.

**Questions For The Authors:**

- According to Section 3.3, the context quality of each sample (instance) also needs to be considered. Have you tried to address the context quality of each sample by reflecting the retrieval scores of documents (a hint of the number of relevant documents) in equation 3?

**Reasons To Accept:**

- The writing is clear and neat, and the figures and tables are well placed to help understand the content.

- This paper analyzed the influence of the quality and quantity of context in the retrieval augmented model, which is helpful to the community as it can be used as a guideline in other future studies or provide insights.

**Reasons To Reject:**

- (Minor) In my opinion, the analysis in Section 3.4.1 is actually quite obvious and not very meaningful. When there are multiple relevant documents, attention probabilities are inevitably divided for each document, so it will show a long-tailed distribution compared to a setting with only a single relevant document.

I do not fully agree with some of the experimental settings. If I've misunderstood, I hope the authors deal with this during the response period, and if they do, I'll be happy to revise my score.
- Given the experimental results in Section 3.1, it seems more accurate that the performance degradation is due to differences in the ability/need to handle multiple related documents rather than differences in context quality (line 260). In the center and right-hand side experiment results of Figure 1, there was only one relevant document at the time of inference. Therefore, it can be interpreted that the performance degrades as the context quality increases because the model learned to handle multiple relevant documents during training. To prove that the performance is poor due to the difference in the context quality itself as the authors argue in line 260, it should be seen that the performance drops when the context quality increases above 0.3 with n_train = 10 on the left-hand side of figure 1 (where n+_eval > 1). In the absence of such evidence, the claim should be reduced to single/multiple passage.

**Reproducibility:**

5: Could easily reproduce the results.

**Reviewer Confidence:**

4: Quite sure. I tried to check the important points carefully. It's unlikely, though conceivable, that I missed something that should affect my ratings.

---

> ### Author Rebuttal · Authors · 2023-08-28
>
> **Comments for Reasons To Reject section**
>
> > (Minor) In my opinion, the analysis in Section 3.4.1 is actually quite obvious and not very meaningful. When there are multiple relevant documents, attention probabilities are inevitably divided for each document, so it will show a long-tailed distribution compared to a setting with only a single relevant document.
>
> We apologize for any confusion that may have arisen due to the lack of explicit explanation in the paper, and we’d like to clarify our analysis.
>
> In lines 389 and 395, when we compare the attention distributions, we compare distributions shown within the same panel in Fig. 3 (e.g., the blue dashed line and the red solid line). In each panel of Fig. 3, we controlled and fixed the number of relevant documents during the evaluation. The results demonstrate that even under the same evaluation conditions (i.e., the same number of relevant documents), models with a lower training context quality show more long-tailed distributions to relevant passages.
>
> > Given the experimental results in Section 3.1, it seems more accurate that the performance degradation is due to differences in the ability/need to handle multiple related documents rather than differences in context quality (line 260). ... In the absence of such evidence, the claim should be reduced to single/multiple passage.
>
> In Fig. 7 in the appendix, we have provided additional experimental results under different inference conditions that we consider relevant to your concerns. According to the results in Fig. 7, the performance degrades in both cases when the number of relevant documents during inference is 1 and 3 if the context quality is higher than that used during training (represented by the red dashed line). In particular, focusing on the results represented by the blue dotted line in the bottom-right panel of Fig. 7, the model's performance deteriorates when the number of relevant documents during training increases from 1 to 3, even though the number of relevant documents during inference is 3. Based on these results, we consider that our claim holds regardless of the number of relevant documents.
>
> **Comments for Questions For The Authors section**
>
> > According to Section 3.3, the context quality of each sample (instance) also needs to be considered. Have you tried to address the context quality of each sample by reflecting the retrieval scores of documents (a hint of the number of relevant documents) in equation 3?
>
> We have not conducted experiments to estimate the context quality for each sample (i.e., each question) using relevance scores (e.g., retrieval scores or FiD attention scores) and dynamically change the intervention on cross-attention (e.g., temperature parameter) for each sample. This remains an area for future work. However, in preliminary experiments, we analyzed the correlation between passage quality (i.e., how much the passage contributed positively to correctly answering the question) and cosine similarity between a question and a passage embedding, and we did not observe strong correlations. The result suggests that using relevance scores to accurately classify relevant documents and estimate their numbers might not be straightforward. Our proposed method avoids the necessity of accurately estimating quality for each sample by estimating the single temperature parameter for the entire dataset using the development set.
>
> Note: Considering the result in the bottom panels of Fig. 5, it seems possible to estimate optimal temperature if we know correct quality during training and inference. Thus, passage quality estimation would be important direction of future work to achieve automatic temperature tuning.

---

### Official Review · Reviewer_mQfV · 2023-08-05

**Soundness:** 4

**Excitement:**

3: Ambivalent: It has merits (e.g., it reports state-of-the-art results, the idea is nice), but there are key weaknesses (e.g., it describes incremental work), and it can significantly benefit from another round of revision. However, I won't object to accepting it if my co-reviewers champion it.

**Paper Topic And Main Contributions:**

The paper studies the effect of context quantity and quality of input documents for FiD model when training for question answering.

**Questions For The Authors:**

A. How do you claim the proposed hyper-parameter, i.e., temperature is better than the others, e.g., number of documents, or different sets of documents selected by heuristics like estimated relevant scores?

**Reasons To Accept:**

This paper provide insights of the effects of context quantity and quality of a retrieval augmented model. The goal of this study is well presented and the experiment designs are well aligned with the goal. I like the layout of the experiments. The analysis of cross-attention probability is interesting and comprehensive.

**Reasons To Reject:**

The novelty of the proposed method, i.e., introducing a hyper-parameter temperature, could be limited. The reason is there are existing hyper-parameters that we can tune when choosing supporting documents, such as number of documents, different sets of documents selected by heuristics like estimated relevant scores. So far, I did not see why temperature is a more advanced hyper-parameter. It would be more significant if the paper could propose an approach to automatically estimate this temperature without tuning it by cross-validation.

**Reproducibility:**

4: Could mostly reproduce the results, but there may be some variation because of sample variance or minor variations in their interpretation of the protocol or method.

**Reviewer Confidence:**

3: Pretty sure, but there's a chance I missed something. Although I have a good feel for this area in general, I did not carefully check the paper's details, e.g., the math, experimental design, or novelty.

**Typos Grammar Style And Presentation Improvements:**

Line 395: Contray -> On the contrary

Figure 5 is a bit difficult to understand.

---

> ### Author Rebuttal · Authors · 2023-08-28
>
> > Questions For The Authors:
> A. How do you claim the proposed hyper-parameter, i.e., temperature is better than the others, e.g., number of documents, or different sets of documents selected by heuristics like estimated relevant scores?
>
> The proposed temperature parameter has distinct characteristics from existing hyperparameters and works complementary rather than competitively with the existing hyperparameters.
>
> Existing hyperparameters, such as the number of documents and heuristical selection based on relevance scores, affect model performance by chainging the set of input passages itself. Since a trade-off exists when selecting input passages (e.g., between recall and quality), tuning these hyperparameters is important.
>
> Nevertheless, relying solely on these hyperparameters has its drawbacks. As shown in our paper, not only context itself but also a quality gap between training and inference time influences the performance of FiD. A larger quality gap can lead to performance degradation. Therefore, unless reducing the quality gap via a costly retraining of FiD, there is a risk of underestimation of the model performance on a specific hyperparameter setting that leads to a larger quality gap, leading to suboptimally tuned hyperparameters.
>
> Contrary to the existing hyperparameters, our proposed temperature parameter does not change input contexts to reduce the effect of the quality gap. Therefore, it can not only improve the performance on a given fixed context during inference but also reduce the aforementioned underestimation effect due to the quality gap without any retraining.
>
> In conclusion, the temperature parameter works complementarily to the existing hyperparameters that adjust context itself, thereby improving their tuning performance.
>
> > It would be more significant if the paper could propose an approach to automatically estimate this temperature without tuning it by cross-validation.
>
> We acknowledge the importance of automatically adjusting the temperature parameter, as you pointed out, and consider it a crucial direction for future research.

---

### Meta-Review · Area_Chair_mQNh · 2023-09-19

**Recommendation:** 3

**Metareview:**

This paper provides an interesting analysis of the effects of context quality (ie the proportion of relevant passages in the top N documents)  on extractive open-domain QA. They found that FiD's resulting performance is strongly affected by the quality of the context during training and can overfit the training data distributions, resulting in degraded performance when test context quality differs. Based on the observations, they introduce a temperature parameter to the loss, to change the sharpness.

Overall, the research questions and experiments are intriguing, but the main takeaway isn't really surprising to me. To summarize, the core finding (e.g., Section 3.1; *For a given evaluation context quality, models trained with similar context quality showed the highest performance*) is not surprising and can be explained by the tendency of a model performing poorly when there's distribution shift between training and test.  This point is also noted by the reviewer t9yE. On the other hand, I think some findings on attention distributions are somewhat underexplored in prior work and provide unique contributions, and may inspire future work to further investigate some mixed results in this work (e.g., Table 2).

Another concern I have is that the analysis is limited to a single model architecture (FiD) on an extractive question-answering task, and given the experimental designs, this analysis may not be easily adapted to other architecture or tasks.

---

### Decision · Program_Chairs · 2023-10-07

**Decision:**

Accept-Findings

**Comment:**

This paper provides an interesting analysis of the effects of context quality (ie the proportion of relevant passages in the top N documents)  on extractive open-domain QA. They found that FiD's resulting performance is strongly affected by the quality of the context during training and can overfit the training data distributions, resulting in degraded performance when test context quality differs. Based on the observations, they introduce a temperature parameter to the loss, to change the sharpness.

Overall, the research questions and experiments are intriguing, but the main takeaway isn't really surprising to me. To summarize, the core finding (e.g., Section 3.1; *For a given evaluation context quality, models trained with similar context quality showed the highest performance*) is not surprising and can be explained by the tendency of a model performing poorly when there's distribution shift between training and test.  This point is also noted by the reviewer t9yE. On the other hand, I think some findings on attention distributions are somewhat underexplored in prior work and provide unique contributions, and may inspire future work to further investigate some mixed results in this work (e.g., Table 2).

Another concern I have is that the analysis is limited to a single model architecture (FiD) on an extractive question-answering task, and given the experimental designs, this analysis may not be easily adapted to other architecture or tasks.